# University Academic Performance Development Prediction Based on TDA

**DOI:** 10.3390/e25010024

**Published:** 2022-12-23

**Authors:** Daohua Yu, Xin Zhou, Yu Pan, Zhendong Niu, Xu Yuan, Huafei Sun

**Affiliations:** 1School of Computer Science and Technology, Beijing Institute of Technology, Beijing 100081, China; 2School of Mathematics and Statistics, Beijing Institute of Technology, Beijing 100081, China; 3School of Computing and Information, University of Pittsburgh, Pittsburgh, PA 15260, USA; 4Yangtze Delta Region Academy of Beijing Institute of Technology, Jiaxing 314019, China

**Keywords:** topological data analysis, short time series analysis, Markov chain, university academic performance

## Abstract

With the rapid development of higher education, the evaluation of the academic growth potential of universities has received extensive attention from scholars and educational administrators. Although the number of papers on university academic evaluation is increasing, few scholars have conducted research on the changing trend of university academic performance. Because traditional statistical methods and deep learning techniques have proven to be incapable of handling short time series data well, this paper proposes to adopt topological data analysis (TDA) to extract specified features from short time series data and then construct the model for the prediction of trend of university academic performance. The performance of the proposed method is evaluated by experiments on a real-world university academic performance dataset. By comparing the prediction results given by the Markov chain as well as SVM on the original data and TDA statistics, respectively, we demonstrate that the data generated by TDA methods can help construct very discriminative models and have a great advantage over the traditional models. In addition, this paper gives the prediction results as a reference, which provides a new perspective for the development evaluation of the academic performance of colleges and universities.

## 1. Introduction

Academic performance is crucial for evaluating the level of universities. In the mainstream university leaderboards, the academic performance of a university is usually quantified as various statistical indicators, e.g., the number of published papers, the amount of research funding and so on. Our previous work [1] has researched the effects of different academic indicators and proposed a new evaluation method of the university academic level based on statistical manifolds. In addition, we have conducted studies on the academic growing potential of individuals [2]. During our research, we noticed that although there had been quite a lot of work on the design of evaluation criteria for academic level rating in a period [3,4,5,6], not so much attention has been paid to the analysis of academic growth potential. In other words, previous work only focused on the academic level comparison among different universities but lacked the excavation of academic-level development with time for the single school. As a matter of fact, the academic growing potential can serve as the basis of policies as well as one more reference for university evaluation, just as what trend analysis can do in the fields of finance, energy, and other industries. The academic development can be represented by the variation trend of specified statistical indicators, which is the main research object of this article.

As a matter of fact, the study of variation patterns of university academic indicators is a typical problem of short time series data analysis. Time series data is a group of sampled sequential data points from a continuous process over time. The analysis of time series data, especially the short one, has been considered one of the most challenging problems in the area of data mining [7]. The first challenge is that it cannot be certain that a piece of time series data consists of enough information to fully describe the real-world process. That is why it is thought that financial markets cannot be predicted [8]. The second, time series data, is often nonstationary, which indicates that the statistics of the time series data, such as mean, variance, and so on, change over time. This requires extra techniques or input data to solve the problem correctly. Moreover, as a sampling of real-world processes, the time series data inevitably contains much noise and often has high dimensionality. These all add up to the difficulties of time series analysis. University academic indicators are usually recorded every year, but the work of recording does not have a long history, and hence the available data is still limited. This may explain why there are hardly any related researches.

Being challenging yet promising, research on approaches for time series data analysis have been active for decades [9]. Traditional approaches mainly focus on fitting the time series data on known models, such as the linear dynamical model [10], regressive model [11], hidden Markov model [12] and ARIMA model [13]. With the development of computing power and neural networks theory, nowadays methods based on deep learning are popular and obtain state-of-the-art results in various tasks [14,15]. Our previous work has also gained satisfying prediction models of deep learning [16]. Unluckily, both traditional and modern methods cannot achieve satisfying results on short time series data. Traditional methods cannot correctly give the results when data consists of much noise, which is common for time series data. Furthermore, deep learning methods require data with enough length to extract features; otherwise, it has even worse performance than purely statistical methods [17].

As an emerging area for complicated data processing, topological data analysis (TDA) is an overlap between mathematics and computer science and has been used in biology [18,19], robotics [20,21], finance [22,23], etc. In recent years, TDA for time series data analysis has been growing quickly, and one of the promising methods is persistent homology. By applying persistent homology on data clouds, persistence diagrams can be produced and considerable features can be provided. Previous work has proved the potential of persistent homology in extracting features for time series [24], yet no research on short time series has been published.

To address the problem of university academic indicator prediction, this paper proposes to use TDA, or persistent homology exactly, as the feature extractor to reveal the time series variation patterns. Then, support vector machine (SVM) is used as a classifier to judge the variation trend of indicators. The simulation results show advantage over the classic method Markov chain. By comparing with the traditional model Markov chain, our work proves the efficiency of persistent homology in processing short time series data and capturing variation features. Moreover, by applying the model, we give the prediction of academic indicators of the top universities in mainland China, which could be a reference for other academic evaluation researches.

The paper is organized as follows. In Section 2, we introduce the mathematical basis of TDA, including simplexes and the idea of persistent homology. We also describe our data processing strategies and make necessary validation from the statistical perspective. In Section 3, we first give an overview of the Markov chain, and then perform simulations and give results as the baseline of prediction. In Section 4, we simply give an overview of previous work applying TDA and then describe the simulation and results of using persistent homology.

## 2. Preliminary

Topological data analysis (TDA) is an emerging and rapidly developing field that provides a set of new topological and geometric tools to infer relevant features of potentially complex data. In this section, we briefly introduce some mathematical foundations of TDA and data preprocessing.

### 2.1. Simplicial Homology

Now, we first introduce the related concept of simplicial homology, which is the basis of persistent homology.

The natural domain of definition for simplicial homology is a class of spaces we call Δ-complexes, which are a mild generalization of the more classical notion of a simplicial complex [25].

**Definition 1.** *A* Δ*-complex structure on a space X is a collection of maps*
(1)$${\left\{\sigma_{\alpha }:\Delta^{n(\alpha )}\to X,n\left(\alpha \right)\in Z\geq 0\right\}}_{\alpha \in J}$$
*where $\Delta^{n}$ is a standard n-complex, such that*
*(i)* 
*the restriction $\sigma_{\alpha }|\overset{˚}{\Delta^{n}}$ is injective, and each point of X is in the image of exactly one such restriction $\sigma_{\alpha }|\overset{˚}{\Delta^{n}}$, where the open simplex $\overset{˚}{\Delta^{n}}$ is $\Delta^{n}-\partial \Delta^{n}$, the interior of $\Delta^{n}$;*
*(ii)* 
*each restriction of $\sigma_{\alpha }$ to a face of $\Delta^{n}$ is one of the maps $\sigma_{\beta }:\Delta^{n-1}\to X$. Here, we are identifying the face of $\Delta^{n}$ with $\Delta^{n-1}$ by the canonical linear homeomorphism between them that preserves the ordering of the vertices; and*
*(iii)* 
*a set A⊂X is open iff $\sigma_{\alpha }^{-1}\left(A\right)$ is open in $\Delta^{n}$ for each $\sigma_{\alpha }$.*



**Definition 2.** 
*The simplicial chain group of X is defined as*

(2)
$$\Delta_{n}\left(X\right)=\underset{\alpha ,n(\alpha )=n}{Z}\sigma_{\alpha }=\left\{\sum_{\alpha ,n(\alpha )=n}\lambda_{\alpha }\sigma_{\alpha }|\lambda_{\alpha }\in Z\right\}$$

*where $\lambda_{\alpha }$ are almost all zero.*


**Definition 3.** 
*Define the chain map (boundary homomorphism)*

(3)
$$\partial_{n}:\Delta_{n}\left(X\right)\to \Delta_{n-1}\left(X\right)$$

*via α such that n(α)=n and $\partial_{n}\left(\sigma_{\alpha }\right)=\sum_{i}{\left(-1\right)}^{i}\sigma_{\alpha }|\left[v_{0},\cdots ,{\hat{v}}_{i},\cdots ,v_{n}\right]$, where the hat symbol ^ over $v_{i}$ indicates that this vertex is deleted from the sequence $v_{0},\cdots ,v_{n}$.*


**Remark 1.** 
*By direct calculation, we can see that $\partial_{n}\circ \partial_{n+1}=0$.*


With the above preparations, we can give the definition of the simplicial homology group of *X*.

**Definition 4.** 
*The n-th simplicial homology group of X is defined as*

(4)
HnΔ(X)=Ker∂n/Im∂n+1



The dimension of $H_{n}^{\Delta }\left(X\right)$ is called the *n*-th Betty number. Simplicial homology groups and Betty numbers are topological invariants. A Betty number can represent some topological properties of topological spaces. For instance, the 0-th Betty number counts the connected components, the 1-th Betty number represents the number of holes and the 2-th Betty number computes the numbers of voids.

### 2.2. Persistent Homology

Persistent homology is a method in TDA that can efficiently study the topological features of simplicial complexes and topological spaces. It lets us leave our data in the original high-dimensional space and tells us how many clusters are in the data, and how many looplike structures there are in the data, all without being able to actually see it. The idea of persistent homology is to observe how the simplicial homology changes during a given filteration [26,27].

**Definition 5.** 
*Given dimension n, if there is an inclusion map i of one topological space X to another Y, then it induces an inclusion map on the n-dimensional simplicial chain groups*

(5)
$$i:\Delta_{n}\left(X\right)\to \Delta_{n}\left(Y\right)$$


*Furthermore, this extends to a homomorphism on simplicial homology group*

(6)
$$i_{*}:H_{n}^{\Delta }\left(X\right)\to H_{n}^{\Delta }\left(Y\right)$$

*where $i_{*}$ sends $\left[c\right]\in H_{n}^{\Delta }\left(X\right)$ to the class in $H_{n}^{\Delta }\left(Y\right)$.*


**Definition 6.** 
*A filtration of a simplicial complex K is a nested family of subcomplexes ${\left(K_{r}\right)}_{r\in T}$, where T⊆R, such that for any $r,r^{\prime }\in T$, if $r\leq r^{\prime }$ then $K_{r}\subseteq K_{r^{\prime }}$, and $K=\cup_{r\in T}K_{r}$. The subset T may be either finite or infinite. More generally, a filtration of a topological space M is a nested family of subspaces ${\left(M_{r}\right)}_{r\in T}$, where T⊆R, such that for any $r,r^{\prime }\in T$, if $r\leq r^{\prime }$, then $M_{r}\subseteq M_{r^{\prime }}$ and $M=\cup_{r\in T}M_{r}$.*


For applying persistent homology in a point cloud *P*, there are the following steps.

Step 1: Convert point cloud *P* to a topological space.

Here, we use *VR* complex. For given r≥0 and metric *d* in *P*, the *VR* complex VR(P,r) is the topological space containing different dimensional simplex whose maximum distance among vertices is less than or equal to 2r.

Step 2: Construct a filtration of topological spaces.

A filtration $X_{1}\subseteq X_{2}\subseteq \cdots \subseteq X_{m}$ induces a sequence of homomorphisms on the simplicial homology groups
(7)$$H_{n}^{\Delta }\left(X_{1}\right)\to H_{n}^{\Delta }\left(X_{2}\right)\to \cdots \to H_{n}^{\Delta }\left(X_{m}\right)$$

A class $\left[c\right]\in H_{n}^{\Delta }\left(X_{i}\right)$ is said to be born at *i* if it is not in $i\left(H_{n}^{\Delta }\left(X_{i-1}\right)\right)$. The same class dies at *j* if $\left[c\right]\neq 0\in H_{n}^{\Delta }\left(X_{j-1}\right)$, but $\left[c\right]=0\in H_{n}^{\Delta }\left(X_{j}\right)$.

Step 3: Obtain the resulting information.

Given a filtration Filt $={\left(F_{r}\right)}_{r\in T}$ of a topological space, the homology of $F_{r}$ changes as *r* increases. New connected components can appear, existing components can merge, loops and cavities can appear or be filled, etc.. Persistent homology tracks these changes, identifies the appearing features and associates a lifetime to them. We mark a point in $R^{2}$ at (i,j) if one class is born at *i* and dies at *j*. Hence, we can obtain a persistence diagram by its collection of off-diagonal points
(8)$$D=\left\{\left(b_{1},d_{1}\right),\cdots ,\left(b_{k},d_{k}\right)\right\}$$

Figure 1 is an example of a persistence diagram.

The lifetime or barcode of a point x=(b,d) in *D* is given by pers(x)=|b−d|. The collection of all barcodes is called persistence. The persistence of a dataset contains important topological information about its intrinsic space. In one persistence, long barcodes are interpreted as true topological features of the intrinsic space, whereas short barcodes are interpreted as topological noise. The quantitative discussion of length can be found in [28].

More details on persistent homology can be found in reference [29].

### 2.3. Data Description and Preprocessing

The data used in this paper is provided by the CNKI analysis platform of Chinese university academic achievements [30]. We select the top 50 Chinese mainland universities in terms of scientific research funding in 2021. The names and abbreviations of the 50 universities are listed in Table 1. For each university, we collect six types of its academic indicators from 2010 to 2019, i.e., the number of published papers of SCI and SSCI, the number of state-level funds, the amount of National Natural Science funds, and the number of applicated and authorized patents. We choose these indicators because they are strictly produced and recorded once a year, and they can comprehensively represent the academic level of universities.

An important issue for conventional time series data analysis is the validation of stationarity. A stationary time series is one in which unconditional joint probability distribution does not change over time. Stationarity validation is necessary because many statistical models assume that time series data is stationary, and analysis on nonstationary time series data could result in spurious regression, which means the time series has no relationship with the predicted trend.

One of the popular approaches for stationarity validation is the unit root test (URT) [31]. The null hypothesis of URT is that the unit root exists, i.e., the time series is nonstationary. We choose augmented Dickey–Fuller (ADF) test, which is one of the broadly used methods for URT, to validate the stationarity of our data, i.e., the six categories of academic indicators from 2010 to 2019 of the 50 universities. The implementation is provided by Python API statsmodels.tsa.stattools.adfuller. The API reads the time series data and returns the *p*-value, which is the confidence of accepting the null hypothesis of URT. The result of ADF test on the original data is displayed in Figure 2. We can see that most of the samples have a *p*-value that supports the null hypothesis; hence, we cannot directly use the raw data for analysis.

To address the problem of nonstationarity, we propose to convert time series into its chain indexes, which is a technique usually used in economics [32]. The *n*-th chain index $C_{n}$ is defined as $C_{n}=\frac{D_{n}}{D_{n-1}}$, in which $D_{n}$ is the *n*-th raw data point. An example is given in Table 2.

For our data, every time series sequence contains 10 points. We calculate the chain indexes for each sequence respectively and then perform ADF test on the chain index sequence. The result is shown in Figure 3. We can see that the processed data mostly meets the requirement of time series analysis, and only about 30 samples have *p*-value bigger than 0.1, which are excluded to ensure the whole dataset is stationary.

## 3. Prediction Based on Markov Chain

### 3.1. Overview of Markov Chain

The Markov chain (MC) can be said to be the cornerstone of machine learning and artificial intelligence, and has a wide range of applications in finance [32], weather forecasting [33], and many other fields. In fact, a Markov chain is a special kind of stochastic process where the next state of the system depends only on the current state and not on the previous ones.

**Definition 7.** 
*Stochastic process in form of discrete sequence of random variables $\left\{X_{n}\right\},n=1,2,\cdots$ is said to have the Markov property if Equation (Equation 9) holds for any finite n, where particular realizations $x_{n}$ belong to discrete state space $S=\left\{s_{i}\right\},i=1,2,\cdots ,k$. We have*

(9)
$$\begin{array}{c} P\left(X_{n+1}=x_{n+1}|X_{1}=x_{1},X_{2}=x_{2},\cdots ,X_{n}=x_{n}\right)=P\left(X_{n+1}=x_{n+1}|X_{n}=x_{n}\right) \end{array}$$



Generally, MC is described by vectors p(n) which give unconditional probability distributions of states, and transition probability matrix P which gives conditional probabilities $p_{ij}=P\left(X_{n+1}=s_{j}|X_{n}=s_{i}\right),i,j=1,2,\cdots ,k$ where $p_{ij}$ may depend on *n*. Development of p(n) is given by recurrence Equation (Equation 10), where ${}^{T}$ denotes transposition. We have
(10)$$\begin{array}{c} p{\left(n+1\right)}^{T}=p{\left(n\right)}^{T}P,n=1,2,\cdots \end{array}$$

### 3.2. Simulation and Results

As mentioned in Section 2.3, to ensure the stationarity of time series, the chain indices are used for input data. Considering MC model is meant to predict a sequence of discrete states and chain indices are continuous real numbers, we make projections that map chain indices to some discrete states. We define state spaces $S_{1}$, $S_{2}$, and $S_{3}$ as below. The intervals are divided according to practical demands and the distribution of data. We have
(11)$$\begin{array}{cc} \; & S_{1}=\left\{D,G\right\},D:C_{n}\leq 1,G:C_{n}>1 \end{array}$$
(12)$$\begin{array}{cc} \; & S_{2}=\left\{D,G_{1},G_{2}\right\},D:C_{n}\leq 1,G_{1}:1<C_{n}\leq 1.5,G_{2}>1.5\\ \; & S_{3}=\left\{D,G_{1},G_{2},G_{3}\right\},D:C_{n}\leq 1,G_{1}:1<C_{n}\leq 1.25, \end{array}$$
(13)$$\begin{array}{cc} \; & \;\;\;\;\;\;\;\;G_{2}:1.25<C_{n}\leq 1.5,G_{3}:C_{n}>1.5 \end{array}$$

In the simulation, we truncate every 9-element sequence into an 8-element input sequence and an element to predict. The transition probability matrix *P* is given as
(14)$$\begin{array}{c} p_{ij}=P\left(C_{n+1}=s_{j}|C_{n}=s_{i}\right),s\in S \end{array}$$

After the construction of the transition probability matrix, we can then use the recurrence equation to give predictions. We have
(15)$$\begin{array}{c} p{\left(n+1\right)}^{T}=p{\left(n\right)}^{T}P \end{array}$$

In this paper, we use some classic metrics to evaluate the performance of different models and the related definitions are given briefly as follows.

In binary classification tasks, we can divide samples into positive samples and negative samples. We refer TP to the number of true positive samples classified by the model, and similarly, FN to false negative samples, FP to false positive samples, as well as TN to true negative samples. Moreover, for multiclassification tasks, we can select one specified class as the positive samples and the other as negative samples. On this basis, we can define precision, recall and accuracy as follows:(16)$$\begin{array}{c} \mathrm{precision}=\frac{TP}{TP+FP} \end{array}$$
(17)$$\begin{array}{c} \mathrm{recall}=\frac{TP}{TP+FN} \end{array}$$
(18)$$\begin{array}{c} \mathrm{accuracy}=\frac{TP}{TP+TN+FP+FN} \end{array}$$

In case the model has high precision but low recall or the contrary, F1-score is also introduced. The F${}_{\beta }$-score is defined as
(19)$$\begin{array}{c} F_{\beta }-\mathrm{score}=\left(1+\beta^{2}\right)\times \frac{\mathrm{precision}\times \mathrm{recall}}{(\beta^{2}\times \mathrm{precision})+\mathrm{recall}} \end{array}$$
and the F1-score is most usually used. These four metrics will be used to evaluate the performance of the models. It is worth mentioning that we select *D*-state as the positive samples as there are fewer *D*-state samples and it has higher requirements for the models to give the correct results.

In the simulations of MC, the starting state is directly given by p(1). We use Python to implement the simulation, and the results are shown in Table 3.

The results show that the accuracy and the precision score keep going down with the increase of states, but the recall score goes up. As there are many more growing states ($C_{n}>1$) than decreasing states ($C_{n}\leq 1$), the model can achieve high accuracy as long as it has a bias toward predicting increase. Noticing that the recall score is fairly low at the beginning, we can conclude that the MC model is highly biased and actually cannot make very good predictions. The sequence is too short for the MC model to learn enough probability information.

## 4. Prediction Based On TDA

### 4.1. Overview of TDA

Although one can trace back geometric methods used for data analysis long ago, TDA really started as a field with the pioneering works of Edelsbrunner et al. [34] and Zomorodian and Carlsson [35] in persistent homology and was popularized in a landmark paper by Carlsson [36].

The general purpose of TDA is to extract effective information from high-dimensional data, which belongs to unsupervised learning and representation learning from the perspective of machine learning. Over the past few years, researchers have provided TDA with many efficient data structures and algorithms that are now implemented and available and easy to use through the standard libraries.

In recent years, the number of publications on the application of topological data analysis has increased greatly. Below we list only some of the results, 3D shape analysis by Skraba [37], material science by Kramar [38], multivariate time series analysis by Khasawneh and Munch [20], image analysis by Qaiser [39], and financial investment by Goel [40]. These successful results have demonstrated the effectiveness of topological and geometric approaches. In the next section, we will apply persistent homology to feature generation on data from 50 universities.

### 4.2. Feature Generation with Persistent Homology

As opposed to conventional time series analysis methods, persistent homology takes a data cloud sampled from time series as input; hence, there is no concern about stationarity [41]. As persistent homology relies on a distance metric, we first normalize the raw data to ensure the scales of different indicators are comparable. Then we apply Takens’s embedding to convert time series into data clouds. According to the previous research [23,24,42], we select the delay parameter τ as 1 and the dimension parameter *d* as 3. Hence, the nine-element input sequence is converted into a group of seven points with three dimensions. Then, we can apply persistent homology on the data clouds. As introduced in Section 2.2, the output of persistent homology is a set of pairs of birth times and death times of complexes, which can be presented as persistence diagrams or barcodes. Then, statistics can be produced from the persistence diagrams. The pipeline of TDA can be summarized as Figure 4. In this article, we use the Python package ripser [43] to compute the persistence diagrams.

To explicitly present the output of persistent homology, we select three samples with growing trends and the other three with decreasing trends, and show their persistence diagrams in Figure 5.

We can see that the lifetimes in dimension $H_{0}$ show strong correlations with the trends. The ones with growing trends have smaller maximum lifetimes, and their death times are more dense. This inspires us to solve the statistics of the lifetimes of each diagram and check if they are good features for predicting trends. In $H_{0}$ dimension all points have birth time $t_{b}=0$; hence, lifetime equals death time $t_{d}$. The statistics we used include:sum of lifetimes: $\sum t_{d}$;mean of lifetimes: $\mu (t_{d})$;standard deviation of lifetimes: $\sigma (t_{d})$;maximal lifetime: $M(t_{d})$;minimal lifetime: $m(t_{d})$;number of lifetimes bigger than 0.5$M(t_{d})$: $N_{0.5M}$;number of lifetimes bigger than 0.5$\mu (t_{d})$: $N_{0.5\mu }$.

After obtaining the statistics, we can solve their correlations and the results are presented as Table 4.

We can see that the statistics obtained from persistence diagrams are well correlated with the trends; hence, they are good features used by the downstream algorithm to give predictions. We use PCA to map the time series data into planes to visualize the data distribution before and after persistent homology. The figures are as Figure 6 and Figure 7. We can see that the statistics produced by persistence diagrams actually have a more explicit pattern and are easier for classification.

To further explore how persistent homology acts on the inputs, we apply sensitivity analysis to this process. We choose to use Sobol method, which decomposes the variance of output into fractions and attributes them to the input variants as the direct measures of sensitivity. It is one of the most widely used sensitivity analysis methods, as it can adapt to nonlinear responses and it is a global method, which means it gives sensitivity measures based on the whole input space. The implementation is achieved by using the Python package salib [44]. It provides tools to easily generate input samples according to specified bounds and solve the sensitivity scores by using inputs and outputs of the model. In our simulations, we use the scaled data (as their bounds are easily determined) as inputs and the statistics of persistence diagrams as outputs, which is displayed in Figure 4, and we set the number of samples to 1024. The results of the total sensitivity contributions for the six statistics are displayed in Figure 8. Note that the sum and the mean of lifetimes have the same sensitivity bar plot because the mean is just computed by dividing the sum into the same constant.

From Figure 8, we can discover that the “body” of the input variants has higher sensitivities compared to its “head” and “tail” parts. We attribute this to the use of Takens’ embedding, and this distribution helps persistent homology focus more on the global trends instead of being influenced by local disturbances. In addition, we can find that the statistics with higher linear dependence on the trends overall have a lower sensitivity, which indicates our method does have great robustness. In addition, as a matter of fact, all six statistics are statistically significant under an F-test relative to all the input variants, which again validates that these statistics can reflect the trends and are good features for prediction.

### 4.3. Trend Forecasting with SVM

Support vector machine (SVM) is a very famous supervised machine learning algorithm. The vanilla SVM uses training samples to find a hyperplane that maximizes the minimum distance of different classes in the feature space. Later, with the introduction of kernel methods, people found that SVM performs well for both linear and nonlinear analyses, and can be used for both classification (SVC) and regression (SVR) [45]. In our simulations, we use the statistics solved in Section 4.2 as features to forecast the trends. Three kernels, i.e., the linear kernel, the polynomial kernel and the Gaussian radial basis function (RBF) kernel are respectively applied to better fit the data. Three-quarters of the data is randomly selected as the training dataset to produce an SVM classifier with one of the three kernels, and the rest of the data is used as a test dataset. For each kernel, we conduct 10 simulations, and record the average results. The numerical implementation of SVM is provided by Python package sklearn.svm [46] and we only change the specified kernels, keeping the other parameters default. The results in different state spaces are as Table 5, Table 6 and Table 7. Note that the SVM with polynomial kernel has reported zero values for precision and recall, which indicates that this kernel cannot correctly distinguish the positive samples (*D*-state). In order to make head-to-head quantitative comparisons, we also test the vanilla SVM classifier with the chain-indexed data (to ensure the stationarity) and the corresponding results are also displayed in Table 5, Table 6 and Table 7. Interestingly, when simulating on original data, the vanilla SVM with the linear kernel cannot converge instead of performing well as it does on the TDA statistics.

From the simulation results, we can conclude that statistics from persistent homology prove to be good features for the prediction of variation trends of short time series data. In the three kernels used, the linear kernel performs the best on the TDA statistics, whereas the RBF kernel cannot work properly. This indicates that the statistics have linear relationships with the trend, as the RBF kernel should perform well on nonlinear datasets. In contrast, the nonlinear kernels perform well relatively on the original data, but do not rival the performance on the TDA statistics. This proves that persistent homology is a powerful tool with which to dig the underlying relationships and convert the nonlinear relationships into linear in our simulations. Moreover, the recall and the F1-score keep a high level even with the increase of states when using TDA statistics, which supports the idea that data produced by persistent homology together with SVM can achieve very good predictions.

To bring the university development forecast into full play, we further apply SVC with linear kernel on the top 20 universities to obtain an instructive result. We collected the corresponding data from 2010 to 2021 and use the same simulation strategies as above. We train the model with leave-one-out cross-validation. The prediction results are displayed in Table 8. We can see that the funding indicators show a general decline among more than half of the universities, whereas the publication- and patent-related indicators keep increasing mostly. In addition, we can conclude that, though the overall variation trend of the academic indicators of the top 20 universities appears to be rising, the universities likely to have decreasing indicators mainly are the provincial colleges, and their academic backgrounds are mainly natural science or social science, rather than engineering. This phenomenon can also be validated by our previous work [1], as the universities with the same (decreasing) trends are more likely to be clustered together.

## 5. Conclusions and Future Work

Based on the fact that the prediction of university academic indicator variation trends is hardly studied, this paper proposes to obtain time series patterns by using persistent homology. We use classic TDA pipeline methods to extract features from raw data and SVM to make predictions. The results show that TDA methods have an obvious advantage over the conventional statistical Markov chain method in terms of accuracy and F1-score, which indicates that TDA methods can fully capture the variation patterns. Our work proves the great potential of persistent homology in the field of short time series data analysis. The prediction results also provide a new perspective for evaluating the academic performance development of universities. Compared to the previous work based on conventional statistical and bibliometrics methods [47], our work has a solid foundation of mathematical methodology, and thus can avoid the subjective influence introduced by researchers and can be applied in a wider range of related indicator evaluation.

In the future, we would like to conduct further research on the combination of TDA methods and deep learning. It is also important to address the problem of fitting nonequal-length data to persistent homology methods, as in practice time series data at a specific point can be missing, and the existing TDA methods require sequences of equal length on which to perform transitions. Future work would play a significant role in the practical application of TDA methods. In addition, more studies can be carried on to reveal the relationships between university development and its subject background as well as many other factors. The designing of evaluation methods for combining existing rating system with the growing potential of university level is also a big challenge. In brief, the research of quantitative university evaluation still has a long way to go.

## Figures and Tables

**Figure 1 entropy-25-00024-f001:**
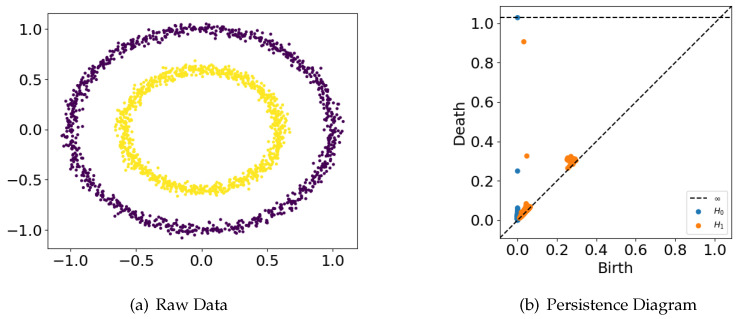
Raw data and its persistence diagram in $H_{0}$ and $H_{1}$. The different colors correspond to different distributions of data and they can be distinguished by the $H_{1}$ persistence diagram.

**Figure 2 entropy-25-00024-f002:**
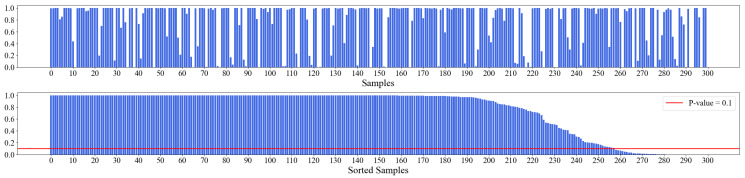
*p*-values of ADF test on raw data. The results show that most of the data is nonstationary.

**Figure 3 entropy-25-00024-f003:**
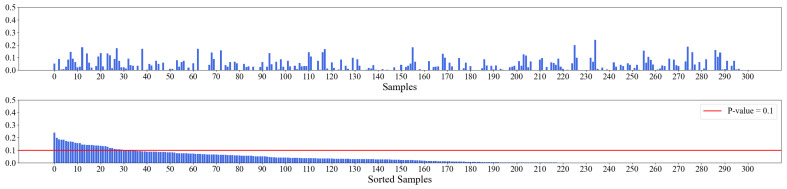
*p*-values of ADF test on chain indexes. Most of the data is stationary after the convertion of chain indices.

**Figure 4 entropy-25-00024-f004:**
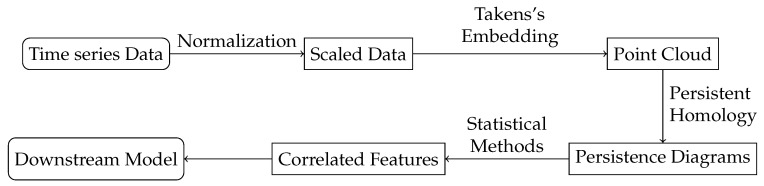
TDA flowchart.

**Figure 5 entropy-25-00024-f005:**
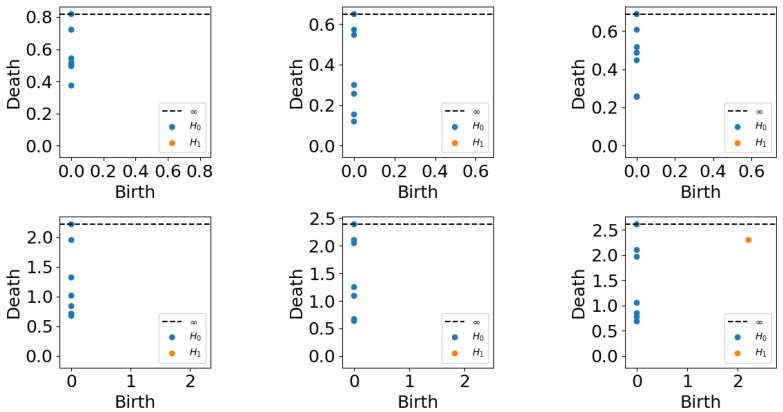
Persistence diagram samples.

**Figure 6 entropy-25-00024-f006:**
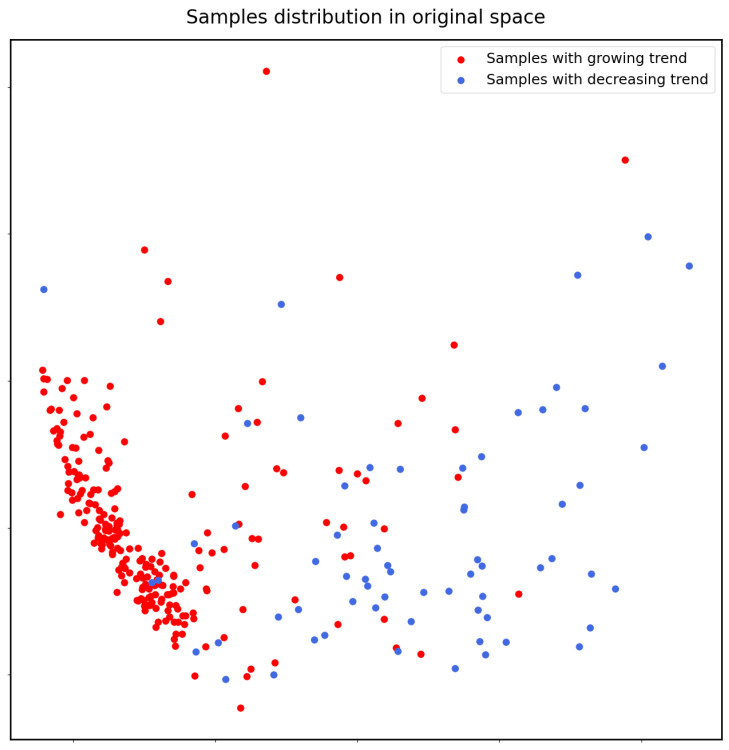
Samples in original space are disordered; hence, it can be hard to give predictions.

**Figure 7 entropy-25-00024-f007:**
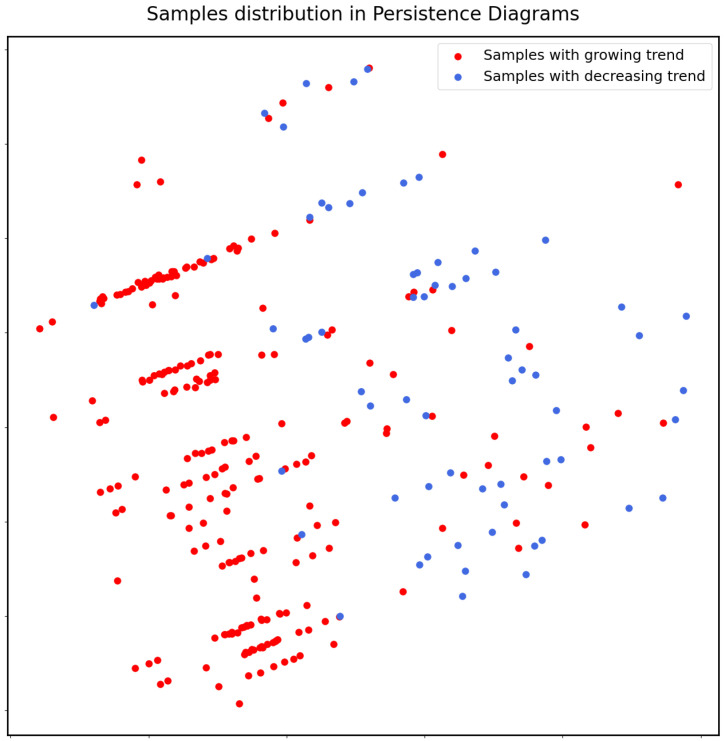
After feature extraction by TDA, samples are arranged by different variation trends, which provides convenience for downstream models.

**Figure 8 entropy-25-00024-f008:**
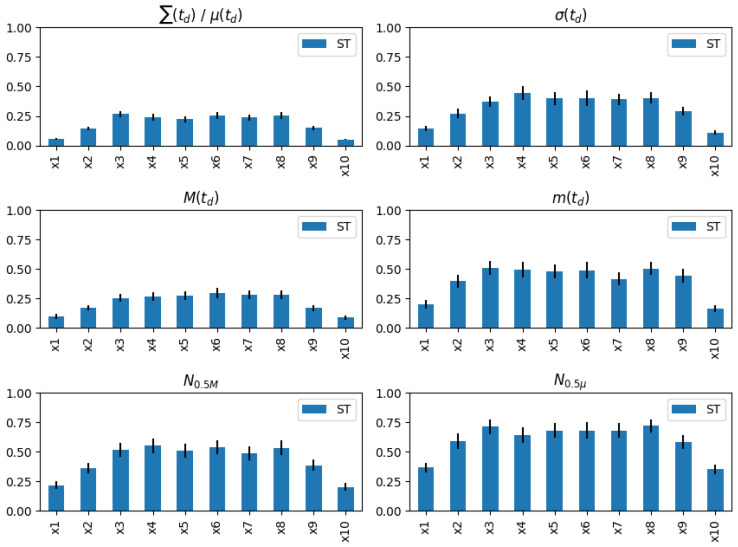
Bar plot of the total sensitivity contributions. The input variants $x_{1}\ldots x_{10}$ correspond to the yearly scaled data and their bounds are determined by the quantile of the original data.

**Table 1 entropy-25-00024-t001:** The names and abbreviations of the 50 universities.

No. 1 to No. 25	No. 26 to No. 50
Tsinghua University (THU)	Hohai University (HHU)
Zhejiang University (ZJU)	Hunan University (HNU)
Peking University (PKU)	East China Normal University (ECNU)
Sun Yat-sen University (SYSU)	South China University of Technology (SCUT)
Shanghai Jiao Tong University (SJTU)	Lanzhou University (LZU)
Fudan University (FDU)	Nanjing University (NJU)
Shandong University (SDU)	Nanjing University of Aeronautics and Astronautics (NUAA)
Huazhong University of Science and Technology (HUST)	Nanjing University of Science and Technology (NJUST)
Xi’an Jiaotong University (XJTU)	Nankai University (NKU)
Southeast University (SEU)	Shenzhen University (SZU)
Beihang University (BHU)	Tianjin Universally (TJU)
Harbin Institute of Technology (HIT)	Wuhan University of Technology (WUT)
Tongji University (TONGJI)	Xidian University (XDU)
Wuhan University (WHU)	Northwest A & F University (NWAFU)
Sichuan University (SCU)	Southwest University (SWU)
Beijing Institute of Technology (BIT)	Southwest Jiao Tong University (SWJTU)
Northwestern Polytechnical University (NPU)	Xiamen University (XMU)
Jilin University (JLU)	Ocean University of China (OUC)
Beijing Normal University (BNU)	University of Science and Technology of China (USTC)
Central South University (CSU)	China University of Mining and Technology (CUMT)
Beijing Jiao Tong University (BJTU)	China Agricultural University (CAU)
University of Science and Technology Beijing (USTB)	Renmin University of China (RUC)
Dalian University of Technology (DUT)	China University of Petroleum-Beijing (CUP)
University of Electronic Science and Technology of China (UESTC)	China University of Petroleum-East China (UPC)
Northeastern University (NEU)	Chongqing University (CQU)

**Table 2 entropy-25-00024-t002:** Chain index example.

Index	0	1	2	3	4	5	6	7	8	9
Raw Data $D_{n}$	1018	1144	1364	1670	2055	2303	2654	2957	3567	4496
Chain Index $C_{n}$		1.12	1.19	1.22	1.23	1.12	1.15	1.11	1.21	1.26

**Table 3 entropy-25-00024-t003:** Results of Markov chain prediction on chain index data.

Metric	State Space $S_{1}$	State Space $S_{2}$	State Space $S_{3}$
Precision	0.600	0.469	0.391
Recall	0.246	0.268	0.321
Accuracy	0.813	0.643	0.497
F1-Score	0.349	0.341	0.353

**Table 4 entropy-25-00024-t004:** Correlations between statistics and trend.

Statistics	$\sum t_{d}$	$\mu (t_{d})$	$\sigma (t_{d})$	$M(t_{d})$	$m(t_{d})$	$N_{0.5M}$	$N_{0.5\mu }$
$\sum t_{d}$	1.00	0.99	0.63	0.86	0.70	0.13	0.25
$\mu (t_{d})$		1.00	0.63	0.86	0.70	0.13	0.25
$\sigma (t_{d})$			1.00	0.89	0.01	−0.52	−0.03
$M(t_{d})$				1.00	0.42	−0.29	−0.02
$m(t_{d})$					1.00	0.45	0.05
$N_{0.5M}$						1.00	0.39
$N_{0.5\mu }$							1.00
**Trend**	−0.62	−0.62	−0.41	−0.60	−0.51	−0.07	−0.11

**Table 5 entropy-25-00024-t005:** Results of different prediction methods on state space $S_{1}$.

Methods	Precision	Recall	Accuracy	F1-Score
Vanilla SVM With Linear Kernel	×	×	×	×
Vanilla SVM With Polynomial Kernel	0.385	0.491	0.760	0.432
Vanilla SVM With RBF Kernel	0.373	0.5	0.747	0.427
PH + SVM With Linear Kernel	0.688	0.846	0.906	0.759
PH + SVM With Polynomial Kernel	0.571	0.677	0.802	0.615
PH + SVM With RBF Kernel	0	0	0.800	0

**Table 6 entropy-25-00024-t006:** Results of different prediction methods on state space $S_{2}$.

Methods	Precision	Recall	Accuracy	F1-Score
Vanilla SVM With Linear Kernel	×	×	×	×
Vanilla SVM With Polynomial Kernel	0.198	0.326	0.587	0.246
Vanilla SVM With RBF Kernel	0.187	0.333	0.560	0.239
PH + SVM With Linear Kernel	0.666	0.677	0.800	0.674
PH + SVM With Polynomial Kernel	0.643	0.529	0.722	0.581
PH + SVM With RBF Kernel	0	0	0.720	0

**Table 7 entropy-25-00024-t007:** Results of different prediction methods on state space $S_{3}$.

Methods	Precision	Recall	Accuracy	F1-Score
Vanilla SVM With Linear Kernel	×	×	×	×
Vanilla SVM With Polynomial Kernel	0.087	0.250	0.347	0.129
Vanilla SVM With RBF Kernel	0.081	0.250	0.320	0.122
PH + SVM With Linear Kernel	0.652	0.747	0.614	0.714
PH + SVM With Polynomial Kernel	0.583	0.636	0.542	0.606
PH + SVM With RBF Kernel	0	0	0.515	0

**Table 8 entropy-25-00024-t008:** Results of indicators variation for top 20 China mainland universities in 2022. “G” represents grow and “D” represents decrease.

University Abbr.	SCI	SSCI	Funds	Fund Amount	Patent App.	Patent Auth.
PKU	D	G	D	D	G	G
BHU	G	G	D	G	G	G
BIT	G	G	D	G	G	G
BNU	G	D	D	D	D	G
SEU	G	G	G	D	D	G
FDU	D	D	G	D	G	G
HIT	G	G	G	G	G	D
HUST	G	G	G	D	G	G
JLU	D	G	D	D	G	G
THU	D	G	G	G	G	G
SDU	G	G	D	D	G	G
SJTU	G	G	G	G	G	G
SCU	D	G	D	D	G	G
TONGJI	G	G	D	D	G	G
WHU	G	G	D	G	G	G
XJTU	G	G	G	G	D	G
NPU	G	D	G	D	G	G
ZJU	G	G	G	G	G	G
CSU	G	G	D	D	G	G
SYSU	G	G	D	D	G	G

## Data Availability

Restrictions apply to the availability of these data. Data was obtained from CNKI and are available at https://usad.cnki.net/ (accessed on 23 February 2022) with the permission of CNKI.

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
