# Peer review of "University Academic Performance Development Prediction Based on TDA"

_entropy, 2022, doi:10.3390/e25010024_

Round 1

Reviewer 1 Report

Paper in interested can be accepted with minor revision

1. Improve abstract with main results.

2. refine your keywords

3. Add details of reproductive number and its sensitivity analysis 

4. add references of the theorem which used in paper

Author Response

Dear Reviewer,

We deeply appreciate the time and efforts you’ve spent in reviewing our manuscript named as “University Academic Performance Development Prediction Based on TDA” with ID: entropy-1998841. According to your suggestions, we have modified our articles and the revises are as follows.

Point 1: Improve abstract with main results.

Response: Your suggestion is appreciated. We have repolished the abstract part and emphasized the main results we obtain.

Point 2: Refine your keywords.

Response: We are grateful to the reviewer for the comments. We have also refined the keywords and they should be more accurate and representative now.

Point 3: Add details of reproductive number and its sensitivity analysis.

Response: Your suggestion is valued by us. We add the sensitivity analysis on the process of Persistent Homology, which is more of a black-box model, and the results show that the sensitivities are actually consistent with the theorem and the selected statistics are all statistical significant under F-test relative to the input variants.

Point 4: Add references of the theorem which used in paper.

Response: Thanks for your attention. We have revised the article and added the missing references of the theorem and other works (like the implementation of algorithms used in simulations). And the references of the theorem about Complexes and Persistent Homology are mainly cited in Chapter 2 (including ref 26-30).

Reviewer 2 Report

The major issue with this paper is that there appears to be no "head-to-head" quantitative comparison of the methods detailed in the work. That is, shouldn't there be a side by side comparison of at least SVM and TDA approaches? This is necessary to prove the efficacy of the TDA approach.

The next most important note: some details about the numerical implementation of SVM and the persistence computation is needed. Are you using a package? If so, which package?

Some minor notes:

1. The sentence "Since... well." in the abstract is incomplete.

2. "Crutial" should be "crucial" -- please be sure to spell check.

3. Line 136: "timeseries" should be "time series"

Author Response

Dear Reviewer,

    We deeply appreciate the time and efforts you’ve spent in reviewing our manuscript named as “University Academic Performance Development Prediction Based on TDA” with ID: entropy-1998841. According to your suggestions, we have modified our articles and the revises are as follows.

Point 1: The major issue with this paper is that there appears to be no "head-to-head" quantitative comparison of the methods detailed in the work. That is, shouldn't there be a side by side comparison of at least SVM and TDA approaches? This is necessary to prove the efficacy of the TDA approach.

Response 1: We deeply understand your concern and appreciate the suggestion. We have added extra simulations of vanilla SVM in the section 4.2 as you suggest. The results show that the original data cannot be handled by merely SVM very well, especially concerning the non-linear part. As a matter of fact, while the TDA statistics + SVM with linear kernels have remarkable results, only SVM with linear kernels does not even converge. And the overall comparisons also show advantage of TDA methods. This proves the efficacy of TDA methods.

Point 2: The next most important note: some details about the numerical implementation of SVM and the persistence computation is needed. Are you using a package? If so, which package?

Response 2: We apologize for the negligence of such important details. Indeed we are using Python packages to do computations. We use Ripser for persistent diagrams and scikit-learn for SVM implementation. We have added citations of these packages as their requests. You can find our codes in this github link and we shall upload the dataset we used after we get authorized by the data provider.

Point 3: The sentence "Since... well." in the abstract is incomplete.

Response 3: Thank you for pointing out this issue. We have corrected this sentence and refined our abstract part.

Point 4: "Crutial" should be "crucial" -- please be sure to spell check.

Response 4: We are sorry for the vocabulary and spell problems. We have carefully rechecked the used vocabularies and their spells.

Point 5: Line 136: "timeseries" should be "time series".

Response 5:  Thanks for your attention and we have corrected all the misspelled “timeseries” into “time series”.

Round 2

Reviewer 1 Report

Accepted